# Clinical Usefulness of New R-R Interval Analysis Using the Wearable Heart Rate Sensor WHS-1 to Identify Obstructive Sleep Apnea: OSA and RRI Analysis Using a Wearable Heartbeat Sensor

**DOI:** 10.3390/jcm9103359

**Published:** 2020-10-20

**Authors:** Takuo Arikawa, Toshiaki Nakajima, Hiroko Yazawa, Hiroyuki Kaneda, Akiko Haruyama, Syotaro Obi, Hirohisa Amano, Masashi Sakuma, Shigeru Toyoda, Shichiro Abe, Takeshi Tsutsumi, Taishi Matsui, Akio Nakata, Ryo Shinozaki, Masayuki Miyamoto, Teruo Inoue

**Affiliations:** 1Department of Cardiovascular Medicine, Dokkyo Medical University, 880 Kitakobayashi, Mibu, Tochigi 321-0293, Japan; takuoari@dokkyomed.ac.jp (T.A.); hyzw@dokkyomed.ac.jp (H.Y.); hirokane1010@gmail.com (H.K.); hal@dokkyomed.ac.jp (A.H.); syoutarouobi@yahoo.co.jp (S.O.); amano@dokkyomed.ac.jp (H.A.); masakuma@dokkyomed.ac.jp (M.S.); s-toyoda@dokkyomed.ac.jp (S.T.); abenana@dokkyomed.ac.jp (S.A.); inouet@dokkyomed.ac.jp (T.I.); 2Division of Cardiology, Eda Memorial Hospital, Kanagawa 225-0012, Japan; tsu-t@kp.catv.ne.jp; 3Union Tool Co. Ltd., Tokyo 140-0013, Japan; matsuit@uniontool.co.jp (T.M.); nakataa@uniontool.co.jp (A.N.); shinozakir@uniontool.co.jp (R.S.); 4Center of Sleep Medicine, Dokkyo Medical University Hospital, Tochigi 321-0293, Japan; miyamasa@dokkyomed.ac.jp

**Keywords:** obstructive sleep apnea (OSA), cyclic variation of heart rate (CVHR), polysomnography, wearable heartbeat sensor

## Abstract

Obstructive sleep apnea (OSA) is highly associated with cardiovascular diseases, but most patients remain undiagnosed. Cyclic variation of heart rate (CVHR) occurs during the night, and R-R interval (RRI) analysis using a Holter electrocardiogram has been reported to be useful in screening for OSA. We investigated the usefulness of RRI analysis to identify OSA using the wearable heart rate sensor WHS-1 and newly developed algorithm. WHS-1 and polysomnography simultaneously applied to 30 cases of OSA. By using the RRI averages calculated for each time series, tachycardia with CVHR was identified. The ratio of integrated RRIs determined by integrated RRIs during CVHR and over all sleep time were calculated by our newly developed method. The patient was diagnosed as OSA according to the predetermined criteria. It correlated with the apnea hypopnea index and 3% oxygen desaturation index. In the multivariate analysis, it was extracted as a factor defining the apnea hypopnea index (*r* = 0.663, *p* = 0.003) and 3% oxygen saturation index (*r* = 0.637, *p* = 0.008). Twenty-five patients could be identified as OSA. We developed the RRI analysis using the wearable heart rate sensor WHS-1 and a new algorithm, which may become an expeditious and cost-effective screening tool for identifying OSA.

## 1. Introduction

Obstructive sleep apnea (OSA) causes a variety of problems in social life, such as reduced work efficiency due to daytime sleepiness or drowsy driving accidents; it also causes so-called lifestyle-related diseases, including hypertension and diabetes [1,2]. In addition, it frequently not only aggravates, but actually causes, life-threatening diseases such as cardiovascular disease [3,4]. Several studies have shown an association between OSA and myocardial infarction [5,6]. Uncontrolled large observational studies have shown that patients with severe OSA, if left untreated, have an increased cardiovascular morbidity and mortality. Fortunately, the introduction of continuous positive airway pressure therapy (CPAP) improves the sleep-related quality of life and further prevents the development of lifestyle-related diseases [7,8]. CPAP therapy has also been reported to reduce the mortality rate of OSA patients [9]. Early detection and treatment of OSA is thus a crucial issue to prevent cardiovascular disease, but most patients remain undiagnosed and untreated [10,11,12]. 

A large number of studies using predominately white populations estimate the prevalence of obstructive sleep apnea (OSA) at approximately 3% to 4% in men and 2% in women [4,13]. Recent data from the Wisconsin Sleep Cohort estimate that 17% of men and 9% of women aged 50 to 70 years have moderate-to-severe OAS [14,15]. Polysomnography (PSG) is currently performed for the diagnosis of OSA. However, many shortcomings limit its generalization to the home environment, and there are few facilities where it can be performed. During the PSG recording procedure, numerous electrodes are attached to the patients’ bodies, disturbing their normal sleep or even making them unable to fall asleep. Additionally, the PSG device is expensive and time-consuming, which makes it unsuitable for home use [16,17].

Thus, a simpler and cheaper alternative test is required. Several alternative techniques have been developed to simplify the measurement of OSA, including single-channel electrocardiograms (ECGs) and pulse oximetry. Fluctuation of the R-R interval (RRI) is referred to as the cyclic variation of heart rate (CVHR) and consists of bradycardia during apnea followed by abrupt tachycardia on its cessation [18]. Several studies have shown that this pattern can be used to detect OSA. Hayano et al. [19] showed that the automated detection of CVHR from ECGs extracted from all-night PSG, using an algorithm of auto-correlated wave detection with an adaptive threshold, provides a powerful screening tool for moderate-to-severe OSA, even in older subjects and in those with cardiac autonomic dysfunction. The Holter ECG has also been used for screening OSA [20,21,22,23]. However, due to a large number of undiagnosed and untreated patients with OSA, a mobile and wearable apparatus for the automated detection of CVHR on ECGs at home is needed to detect patients with OSA.

Therefore, the aim of the present study is to investigate the usefulness of the newly developed RRI analysis using the mobile and wearable heart rate sensor WHS-1 to identify patients with OSA. 

## 2. Materials and Methods

### 2.1. Subjects and Methods

Thirty patients with OSA (22 males and 8 females, average age 63.6 ± 12.1 years, height 161.3 ± 9.2 cm, weight 68.9 ± 15.2 kg, body mass index (BMI) 26.8 ± 4.8 kg /m^2^) were included in this study, as shown in Table 1. In a prior study using portable PSG with electroencephalography (Sleep Watcher1, Compumedics Ltd., Abbotsford, Australia), the patients had 5 oxygen desaturation events per hour and, therefore, were enrolled for a sleep study to assess OSA. We determined the presence of conventional risk factors such as hypertension, diabetes, dyslipidemia, and current smoking and presented them in Table 1. Nineteen patients (63%) had hypertension, 7 (23%) had diabetes, and 13 (43%) had dyslipidemia. One patient received hemodialysis. One patient had chronic obstructive pulmonary disease. Four patients had chronic heart failure due to heart failure with preserved ejection fraction (HFpEF), hypertensive heart disease, old myocardial infarction, and Takotsubo cardiomyopathy. Two patients had old myocardial infarction. One was a patient after mitral valve replacement, and one was a patient after acute aortic dissection. Some patients had medical treatments, including β-blocking agents, calcium-channel blockers, angiotensin receptor blockers, angiotensin converting enzyme inhibitors, diuretics, statins, and antidiabetic drugs (Table 1). The study was approved by the Ethics Committee of the Dokkyo Medical University (No. 28013), and informed consent was obtained from all participants.

### 2.2. PSG Analysis

Overnight pulse oximetry was performed while the patients were breathing room air under stable conditions as a screening test for sleep-related breathing disorders, as previously described [23,24]. An oxygen saturation monitor (Pulsox-M241, Konica Minolta Sensing Inc., Osaka, Japan) was attached to the left fourth finger to measure oxygen saturation (SpO_2_) and pulse rate from 10 p.m. to 6 a.m. The frequency per hour of the reduction of SpO_2_ by more than 3% (the oxygen desaturation index, ODI) and the lowest SpO_2_ were used as parameters for sleep-related breathing disorders. For the patients who had ≥5 ODI events, portable polysomnography with electroencephalography (Alice 5^®^, Alice 6^®^, Respironics, Inc. Murry Ridge Lane, Murrysville, PA, USA) was performed to assess OSA. The parameters as shown in Figure 1 were analyzed by two experienced technicians who were unaware of the study design. The American Academy of Sleep Medicine (AASM) recommendation (2014) was used for a scoring of hypopnea [25]. We scored a respiratory event as a hypopnea if all of the following criteria were met. (1) The peak signal excursions dropped by ≥30% of the pre-event baseline. (2) The duration of the ≥30% drop in the signal excursion was ≥10 s. (3) There was a ≥3% oxygen desaturation from the pre-event baseline, or the event was associated with an arousal. A reduction greater than 90% with the duration ≥10 s was defined as apnea. The number of apnea or hypopnea events per hour was determined as the apnea-hypopnea index (AHI). OSA was defined as an AHI ≥ 5, based on the recommendation of the American Academy of Sleep Medicine Task Force [26].

### 2.3. WHS-1 Analysis

During PSG recording, the electrode was attached on the left anterior chest, as shown in Figure 2. By using wearable WHS-1 (Figure 1, lower part, Union Tool Co. Ltd., Tokyo, Japan), tachycardia with CVHR was detected and diagnosed as OSA when satisfying the criteria described below. In addition, upper and lower limit filters, moving average filters, and body movement and posture filters for acceleration that can specify posture were applied as described below, and noise-free data recorded in the supine position were analyzed.

### 2.4. Parameter Definition

In patients with OSA, the RRI periodically increases and decreases with apnea. This is called CVHR [18]. It occurs because the prolongation of RRI induced by apnea, and subsequently, the increased heart rate due to breathing during hypoxia, is greater in OSA patients than in non-OSA patients [27]. Figure 3A shows the relationship between RRI and SpO_2_ during OSA in a typical case with OSA. The horizontal axis indicates time, and the vertical axis shows SpO_2_ (%, upper trace) and RRI (ms, lower trace). The periodic changes in SpO_2_ indicate that apnea or hypopnea occurs periodically. This increase and decrease in SpO_2_ parallels the changes in RRI shown in the red circle. This suggests that the search for this periodic fluctuation would be sufficient to detect the apnea state due to OSA. Therefore, we developed the method of searching for RRI fluctuations that have a large “valley” of RRI fluctuation, which is a feature of CVHR. The RRI fluctuations are under the influence of autonomic nerve inputs [19,28]. The frequency spectrum analysis of RRI fluctuations revealed high-frequency components (HF, >0.15 Hz) and low-frequency components (LF, 0.04–0.15 Hz) that were differentially affected by autonomic nerve inputs [29,30,31]. HF are associated with cardiac vagal activity, which corresponds to physiological arrhythmias typified by respiratory sinus arrhythmias [32]. LF are mediated by both the sympathetic and vagal nerve [28]. On the other hand, it has been reported that CVHR in OSA coincides with a cycle of very low frequency components (VLF, 0.008–0.04 Hz) [33]. In order to search the valley of the RRI fluctuation, which is a feature of CVHR, the smoothing of the RRI fluctuation corresponding to the HF component and the baselines for identifying the beginning and end of the valley are needed. RRI fluctuations corresponding to the HF component were smoothed by a moving average over a window of 5 s, named the short average of RRI (SRRI). The baseline was determined by averaging the RRI fluctuations corresponding to the VLF component in a 60-s window, named the long average of RRI (LRRI). Figure 3B shows SRRI (blue line) and LRRI (orange line) obtained from the RRI time series in Figure 3A. The horizontal axis represents time (s), and the vertical axis represents the smoothed RRI (ms). It can be seen that SRRI can smooth the fluctuations of the raw RRI, and if LRRI is used as the baseline, the deep valley of the SRRI shows the large valley of the RRI fluctuation that is characteristic of CVHR (Figure 3B). Figure 3C illustrates a part of SRRI and LRRI when CVHR occurs. If the difference between LRRI and SRRI reflects the depth of the RRI valley, the deep valley of RRI, which is a characteristic of CVHR, can be identified, as shown by the arrow in Figure 3C. In other words, the difference between LRRI and SRRI can be taken as the depth of the valley of the RRI fluctuation, and if this depth is greater than a certain value, it can be identified as a deep valley of the RRI fluctuation, which is characteristic of CVHR. However, the magnitude of the RRI fluctuation depends on the heart rate [34]. Since the magnitude of the increase and decrease in CVHR also depends on the heart rate, it is necessary to normalize the depth of the valley of the RRI fluctuation. Therefore, we normalized the depth of the valley of the RRI fluctuation by LRRI in this study. Furthermore, in accord with the results by Guilleminault et al. [18], it can be estimated that if the RRI average is small, the CVHR period and, subsequently, the valley depth of the RRI fluctuation may be small. Therefore, to reduce the dependence of the RRI fluctuation valley depth on the average value of RRI, the normalized depth of the valley of the RRI fluctuation (VRRI) is defined by multiplying the valley of the RRI fluctuation by 1000 ms (60 bpm) and the reciprocal of LRRI, which is a long-term moving average of the RRI as follows (Equation (1)):(1)$$\mathrm{VRRI}(t)\;=\;\frac{\mathrm{LRRI}(t)\;-\mathrm{SRRI}(t)}{\mathrm{LRRI}(t)}\;\times \;\frac{1000}{\mathrm{LRRI}(t)}$$

If RRI is measured at time t, SRRI (*t*) is calculated from the RRIs contained from t (s) to t − 5 (s), and LRRI (*t*) is calculated from the RRIs contained from t (s) to t − 60 (s). In this way, VRRI is calculated sequentially along with the measurement of RRIs. Figure 3D shows the VRRI calculated from RRIs in Figure 3A. The horizontal axis shows the time, and the vertical axis shows VRRI. It can be seen from this that the increase and decrease with periodicity in RRIs, a feature of CVHR, is further emphasized in VRRI. The window time α (s) is provided to search CVHR from the normalized valley depth VRRI of the RRI fluctuation. When the time derivative *VRRI′ (t)* = 0 and *VRRI′ (t)* > β occur more than twice in that window, we decided that a CVHR had occurred. This search was performed during each RRI measurement. An example of searching CVHR from VRRI in Figure 3D is illustrated in Figure 3E. In this case, it can be seen that there are two valleys of RRI fluctuations that satisfy *VRRI′ (t)* = 0 and VRRI (t) > β in a certain window shown by the section within the blue broken line, which suggests that CVHR occurred. Since the period of CVHR is 25 s to 125 s, α needs to be about 180-s-long to include two or more valleys of the above RRI fluctuations. In this study, we used α = 180 (s) and β = 0.090. In addition, the proportion of time in CVHR during sleep in OSA patients is higher than in non-OSA patients, so we called the sum of RRIs with CVHR detected by our method “integrated RRIs of CVHR” during sleep and calculated the ratio of “integrated RRIs of CVHR” (=the sum of RRIs when CVHR occurred) to total sleep time (=the sum of RRIs during sleep) as follows (Equation (2)). When the ratio is more than the threshold (γ), the subjects are suggested to be OSA-positive. In this study, we used γ = 0.090.
(2)$$\mathrm{ratio}\;\mathrm{of}\;\mathrm{integrated}\;RRIs=\frac{integrated\;RRIs\;of\;CVHR\left(=the\;sum\;of\;RRIs\;when\;CVHR\;occurring\right)}{total\;sleep\;time\left(=the\;sum\;of\;RRIs\;during\;sleep\right)}$$

### 2.5. Exclusion of RRI Outliers

The above method of searching for CVHR may be misjudged if there is an outlier in RRI. The outliers are caused by several pathological factors or artifacts. Pathological factors include arrhythmias such as atrial fibrillation, premature contraction, and block. Artifacts are noises on the electrocardiogram that occur due to poor contact between myoelectricity, power lines, and electrodes [35]. Similar to general electrocardiographs, the WHS-1 measurement is also affected by these artifacts. The outliers are measured as RRIs that markedly deviate from normal RRI fluctuations and should be excluded to reduce false positives. In this study, we apply an RRI outlier exclusion method that uses the RRI measured by the WHS-1 and 3-axis acceleration, as described below. 

### 2.6. RRI Filter

As mentioned above, outliers are measured as values that largely deviate from normal RRI fluctuations. Therefore, we exclude RRIs with tachycardia or myoelectricity artifacts of 300 ms or less and sinus arrest or no signal of 3000 ms or more. Furthermore, in order to exclude outliers caused by arrhythmias and sudden artifacts, RRIs that markedly deviate from the latest average RRI are excluded as follows. The instantaneous heart rate *(IHR(i)*) is calculated, and the average heart rate ($\bar{\mathrm{IHR}}\left(n\right)$) is calculated from the past 8 RRIs, where the difference between adjacent instantaneous heart rates (*IHR*(*i*) − *IHR*(*i* − 1)) is less than p1 (Equation (3)).
(3)$$\bar{IHR}\left(n\right)=\frac{1}{8}\overset{8\;points}{\underset{\begin{array}{c} \left|IHR\left(i\right)-IHR\left(i-1\right)\right|\leq p1\\ i=n-1,\;n-2,\ldots . \end{array}}{\sum }}IHR\left(i\right)$$
IHR(i)=60000/RRI(i)

The time series is represented by i, and the RRI to be evaluated as an outlier is written as RRI (*n*). In conditions with $\left|\mathrm{IHR}\left(n\right)-\bar{\mathrm{IHR}}\left(n\right)\right|>p2$, RRI(*n*) is excluded as an outlier.

The short-term average SRRI is calculated from the RRI contained in 5 s. If RRIs are excluded during this period due to outliers, the number of RRIs used to evaluate the average value decreases, and the expected effect of SRRI cannot be obtained. Therefore, when there is an outlier in RRI to be used in the SRRI calculation, CVHR is not searched. On the other hand, the long-term average LRRI has a larger number of RRIs used in the calculation than SRRI, so even if outliers are excluded, the influence of the outlier is small. Therefore, the LRRI is calculated by excluding outlier RRIs. In this study, we used *p*1 = 10 and *p*2 = 15.

### 2.7. Exclusion of Body Movement

WHS-1 can measure not only RRI but, also, 3-axis acceleration at the same time. When WHS-1 is attached to the anterior chest, as shown in Figure 2, the x-axis of the 3-axis accelerometer shows points from the right arm to the left arm. The y-axis shows points from the foot to the head, and the z-axis shows points from the chest to the back. The square root of the sum of each acceleration squared is 1G with a stationary patient; here, G is defined as the unit of the acceleration of gravity. If a patient rolls over, the square root of the sum of each acceleration squared is more than 1G for the patient’s own acceleration motion. To avoid artifacts caused by the patient’s rolling over, the detection of apnea should be invalidated for 30 s before and after the rolling over occurs. If condition
(4)$$\sqrt{X^{2}\left(t\right)+Y^{2}\left(t\right)+Z^{2}\left(t\right)}\geq 1.3G$$
(Equation (4)) is held, it is estimated that the patient rolled over. Here, *X*(*t*), *Y*(*t*), and *Z*(*t*) are X-, Y-, and Z-axis accelerations at time t, respectively.

### 2.8. Exclusion of Postures Other than the Supine Position

OSA can be classified into positional and nonpositional OSA. Postural OSA is more frequently observed in mild OSA, compared with moderate or severe OSA [36,37]. Moreover, it has a long supine duration among all sleeping hours, and postural OSA occurs only in the supine position [37]. 

As shown in Figure 2, it is possible to determine whether or not the patient is in the supine position based on the axial acceleration measured by the WHS-1 attached to the chest. The Y-axis gravitational acceleration is close to 0 in the supine position, so the following is satisfied (Equation (5)).
(5)Z(t)≥0 and Z(t)>Y(t)

The sleeping posture is estimated from the angle θ*xz* between the *XZ* axes. The value of θ*xz* is determined as follows.

In cases with *X*(*t*) ≥ 0 and *Z*(*t*) ≥ 0 (Equation (6)),
(6)$$\theta_{xz}\left(t\right)=\tan^{-1}\frac{Z\left(t\right)}{X\left(t\right)}\left[\mathrm{rad}\right]$$

In cases with *X*(*t*) < 0 and *Z*(*t*) ≥ 0 (Equation (7)),
(7)$$\theta_{xz}\left(t\right)=\pi +\tan^{-1}\frac{Z\left(t\right)}{X\left(t\right)}\left[\mathrm{rad}\right]$$

In cases with *X*(*t*) < 0 and *Z*(*t*) < 0 (Equation (8)),
(8)$$\theta_{xz}\left(t\right)=\pi +\tan^{-1}\frac{Z\left(t\right)}{X\left(t\right)}\left[\mathrm{rad}\right]$$

In cases with *X*(*t*) ≥ 0 and *Z*(*t*) < 0 (Equation (9)),
(9)$$\theta_{xz}\left(t\right)=2\pi +\tan^{-1}\frac{Z\left(t\right)}{X\left(t\right)}\left[\mathrm{rad}\right]$$

When the value of θ*xz* is satisfied with the following criteria (Equation (10)), we decided that it was the supine position.
(10)$$\frac{\pi }{4}\leq \left|\theta_{xz}\left(t\right)\right|\leq \frac{3\pi }{4}$$

### 2.9. Data Analysis 

Data are presented as mean value ± SD. After testing for normality (Kolmogorov-Smirnov), the comparison of means between groups was analyzed by a two-sided, unpaired Student’s *t*-test in the case of normally distributed parameters or by the Mann-Whitney U Test in the case of non-normally distributed parameters. Associations among parameters were evaluated using Pearson or Spearman correlation coefficients. Multiple linear regression analysis with the ratio of integrated RRIs as the dependent variable was performed to identify independent predictors (AHI or 3% ODI). Age, sex, and BMI were employed as covariates. When the independent or dependent data were not normally distributed, the data were logarithmically transformed to achieve a normal distribution. Logistic regression analysis was used. All analyses were performed using SPSS version 24 (IBM Corp., New York, NY, USA) for Windows. A *p*-value of <0.05 was regarded as significant.

## 3. Results

### 3.1. AHI Value and Influence of Body Position

The total mean AHI of all patients was 43.1 ± 26.1, and mean AHI in the supine position was 53.1 ± 27.1. The 3% ODI and minimum SpO_2_ values were 38.7% ± 26.2%, and 78.7% ± 13.1%, as shown in Table 2. Table 2 (right panel) shows correlations between the total AHI value and various parameters. AHI was not significantly correlated with age across all patients but was positively correlated with BMI (*r* = 0.532, *p* < 0.001). AHI was also positively correlated with 3% ODI (*r* = 0.987, *p* < 0.001) and the arousal index (*r* = 0.800, *p* < 0.001) and negatively correlated with the mean SpO_2_ (*r* = −0.399, *p* = 0.013) and minimum SpO_2_ (*r* = −0.652, *p* < 0.001).

The influence of body position on the AHI value is shown in Figure 4. As shown in Figure 4, the AHI value was changed by the postural position. The total AHI value (43.1 ± 26.1) and AHI in the supine position (53.1 ± 27.1) were significantly higher than that recorded in the right lateral (29.1 ± 28.7, *p* < 0.001) and left lateral decubitus positions (34.1 ± 31.2, *p* < 0.001).

### 3.2. Correlation Matrix between Various Parameters and the Ratio of Integrated RRIs 

The correlations between the ratio of integrated RRIs and various parameters are shown in Table 3 and Figure 5. The ratio of integrated RRIs was not significantly correlated with age (Figure 5A, *r* = 0.321, *p* = 0.060) but tended to increase with BMI (Figure 5B, *r* = 0.358, *p* = 0.052). On the other hand, the ratio of integrated RRIs was positively correlated with AHI (total period, Table 3, *r* = 0.416, *p* = 0.022), AHI in the supine position (Figure 5C, *r* = 0.392, *p* = 0.035), and 3% ODI (Figure 5D, *r* = 0.398, *p* = 0.030). However, no correlations were observed between the ratio of integrated RRIs and mean SpO_2_, minimum SpO_2_, and PLMS index, as shown in Table 3. 

### 3.3. Univariate and Multiple Regression Analysis of the Ratio of Integrated RRIs and AHI or 3% ODI

The regression analysis with the ratio of integrated RRIs as the dependent variables and AHI or 3% ODI as the independent variables were investigated in all of the patients, as shown in Table 4. The univariate regression analysis (Table 4) showed that AHI (β = 0.382, *p* = 0.037) was an independent variable to predict the ratio of integrated RRIs. The multiple regression analysis also showed that AHI (β = 0.663, *p* = 0.003) was an independent determinant of the ratio of integrated RRIs even after adjusting for age, sex, and BMI. Similarly, the multiple regression analysis also showed that 3% ODI (β = 0.637, *p* = 0.008) was an independent determinant of the ratio of integrated RRIs even after adjusting for age, sex, and BMI (Table 4).

### 3.4. OSA Diagnosis Rate by WHS-1 Analysis

The patients were classified into three different groups based on their AHI: mild OSA (5 ≤ AHI < 15), moderate OSA (15 ≤ AHI < 30), and severe OSA (30 ≤ AHI). Across all patients, the diagnostic rate detected by the ratio of integrated RRIs was 25/30 (83%). The diagnostic rate detected by the ratio of integrated RRIs was 16/19 (84%) for patients with severe OSA, 7/9 (78%) for patients with moderate OSA, and 2/2 (100%) for patients with mild OSA.

## 4. Discussion 

The major findings of the present study are as follows: With our newly developed RRI analysis using the wearable heartbeat sensor WHS-1 in patients with OSA, the ratio of integrated RRIs positively correlated with AHI and 3% ODI but not with the BMI. In the multivariate analysis, AHI and 3% ODI were extracted as independent factors to define the ratio of integrated RRIs, even when corrected for age, sex, and BMI. Thus, the RRI analysis using the wearable heartbeat sensor WHS-1 with our newly developed algorithm may become a screening test for identifying OSA.

In patients with OSA, the degree of respiratory disorder is often reduced by changing the posture from the supine position to another position, and sleep posture guidance is one of the effective treatment methods that can be performed easily [38,39]. Some studies have also reported that shifting body position from the supine to the lateral decubitus position improves AHI in patients with OSA, and it is more effective for nonobese patients than for obese patients [40,41]. In fact, the present study showed that AHI at the supine position was significantly higher than that recorded at the right and left lateral decubitus positions in patients with OSA. The wearable WHS-1 used in this study can discriminate the body posture. Therefore, it suffices to determine whether or not the patient is apnea only in the supine position. When WHS-1 is attached to the chest, as shown in Figure 2, the Y-axis gravitational acceleration is close to 0 in the supine position. After the additional criteria can be satisfied, we can identify the patient’s position as supine. Apnea judgment was excluded, except when in the supine position. In the present study, the ratio of integrated RRIs significantly correlated with AHI. Thus, the RRI analysis using a wearable heartbeat sensor WHS-1 with our newly developed algorithm, including the function of selecting a supine posture, may become a convenient screening test for identifying OSA. 

The present study showed that the ratio of integrated RRIs positively correlated with AHI and 3% ODI. The diagnostic rate detected by the ratio of integrated RRIs was 84% in patients with severe OSA and 83% in patients with moderate or mild OSA. However, OSA could not be identified in 16% of the OSA patients by the RRI analysis using this algorithm. Several reasons underlying the false negative results may be proposed. CVHR, which consists of bradycardia during apnea followed by abrupt tachycardia on its cessation, is mediated by the autonomic nervous system [18]. In the present study, five patients received β-blocking agents, and OSA could not be detected by these analyses in two of these patients. In addition, two patients had diabetes mellitus (DM), and one patient was treated with insulin therapy. Thus, failure to detect OSA might be due to the drugs and/or the presence of severe impairment of the cardiac autonomic system. Moreover, the present study used the window time α = 180 (s) and β = 0.090 to evaluate the valley depth VRRI of the RRI fluctuation and the threshold γ = 0.090 to detect OSA. Additional studies using a large number of patients will be required to determine the suitable parameter values of α, β, and γ to detect OSA.

Several alternative techniques such as ECGs have been developed to simplify the detection of OSA. Several studies have reported the usefulness of frequency-domain features of heart rate variability (HRV), which is the fluctuation of RRIs in ECGs, to detect apnea screening [42,43,44]. HRV is widely recognized as a noninvasive method for quantifying activities of the autonomic nervous system (ANS). Since apnea affects the ANS and, thus, alterations in HRV [45,46], patients with apnea may be screened by monitoring HRV during sleep. Roche et al. [21] proposed an increase in the relative power of VLF (0.01 to 0.05 Hz) of the inter-beat interval increment (%VLFI) as a marker for OSA. Among a sample of 150 patients referred to a university hospital for clinically suspected OSA, the authors reported the area under the curve (AUC) of 0.70 for identifying the patients with an AHI ≥ 15 per hour and 64% sensitivity and 69% specificity using %VLFI > 4% as the cutoff threshold. In addition, various types of apnea screening algorithms utilizing the ECG directly have been investigated instead of HRV. For example, ECG-derived respiration (EDR) components have been used [21,47,48,49]. 

Several authors have developed algorithms for automated ECG detection of OSA. Khandoker et al. [22] used a machine-learning technique for the automated recognition of OSA from a wavelet analysis of RRIs and ECG-derived respiratory signal. Hayano et al. [19] also showed that the automated detection of CVHR from RRI on ECGs extracted from all-night PSG, using an algorithm of auto-correlated wave detection (ACAT), provides a powerful ECG-based screening tool for moderate-to-severe OSA. The ACAT algorithm provided the temporal position of each CVHR episode and allowed us to predict the severity of OSA directly from the CVHR index. They showed that the correlation coefficient between the AHI and CVHR index was 0.84 and found a CVHR index threshold of 15 per hour for patients with moderate or higher OSA, with a sensitivity of 83% and a specificity of 88%. However, because this method inputs the RRI data of all sleep times at once, it has a poor real-time property. In addition, since it cannot estimate body movements/postures with the algorithm alone, it is not possible to determine whether the OSA is postural. On the other hand, the algorithm used in the present study can successively input RRI, and it can detect apnea nearly in real time. False positives can also be reduced, because RRI outliers caused by posture and body movement can be excluded using the three-axis acceleration measured by WHS-1. In addition, OSA generated from the estimated posture can distinguish between postural and nonpostural OSA. Therefore, when mild postural OSA is detected, coping therapy for apnea that gives intentional stimulation to the patient or promotes posture change may be helpful. However, the method of detecting OSA only by RRIs may give false positives due to periodic leg movement during sleep (PLMS) [19,50,51], which is a leg movement every few seconds during sleep. This is because RRIs cause fluctuations similar to CVHR during PLMS. In the present study, there were no significant differences between the ratio of integrated RRIs and the PLMS index. However, in order to improve the specificity, a method for detecting PLMS may be needed. 

The Holter ECG has been also used for screening OSA [20,21,22,23]. However, considering the large number of undiagnosed and untreated patients with OSA, a mobile and wearable apparatus is required to detect the ECG parameters, including RRIs in patients with OSA at home. Therefore, we used the mobile and wearable WHS-1 system apparatus to measure RRIs in patients with OSA. This wearable heart rate sensor can easily measure precise RRIs based on the ECG. The proposed apnea screening method can also be easily implemented in mobile computers such as smartphones. Thus, the present study shows the possibility that the mobile and wearable heartbeat sensor WHS-1 with the newly developed algorithm can be wildly applied for screening OSA. 

There are several limitations in our current study. First, we studied a small number of patients with OSA who underwent a PSG sleep study and who could be diagnosed with OSA by PSG findings. We have not applied this method to different populations, particularly those with a low pretest probability of OSA. In addition, given the needs for the screening of OSA in general practice and health promotion, studies in the general population are also very important. Thus, further studies using an open invitation for participation in the study and randomization of patients, including low-risk OSA, are required to clarify the clinical usefulness of our program with a wearable heart rate sensor apparatus. Additionally, the additional studies are required to compare the usefulness for detecting OSA by using the WHS-1 system and other screening tools, such as the sleep questionnaires and clinical assessment of the physician. 

## 5. Conclusions

We developed the RRI analysis using the wearable heart rate sensor WHS-1 and new algorithm, which may become an expeditious and cost-effective screening tool for identifying OSA. 

## Figures and Tables

**Figure 1 jcm-09-03359-f001:**
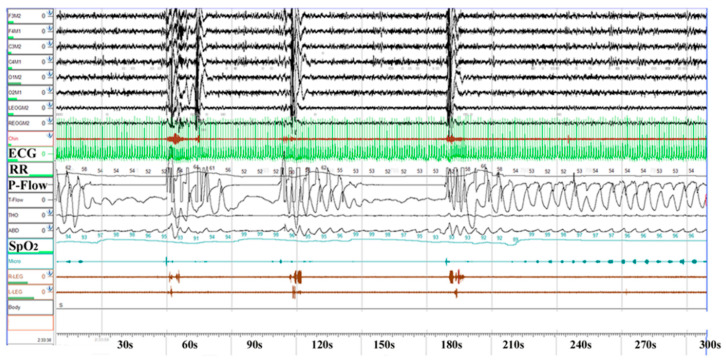
Polysomnography (PSG) findings in a patient with obstructive sleep apnea (OSA). The PSG findings obtained from a 60-year-old patient with severe OSA (apnea-hypopnea index 60) are shown. Note that SpO_2_ repeatedly decreases and then increases with a cyclic variation of heart rate (CVHR). ECG, electrocardiogram; RR, R-R interval.

**Figure 2 jcm-09-03359-f002:**
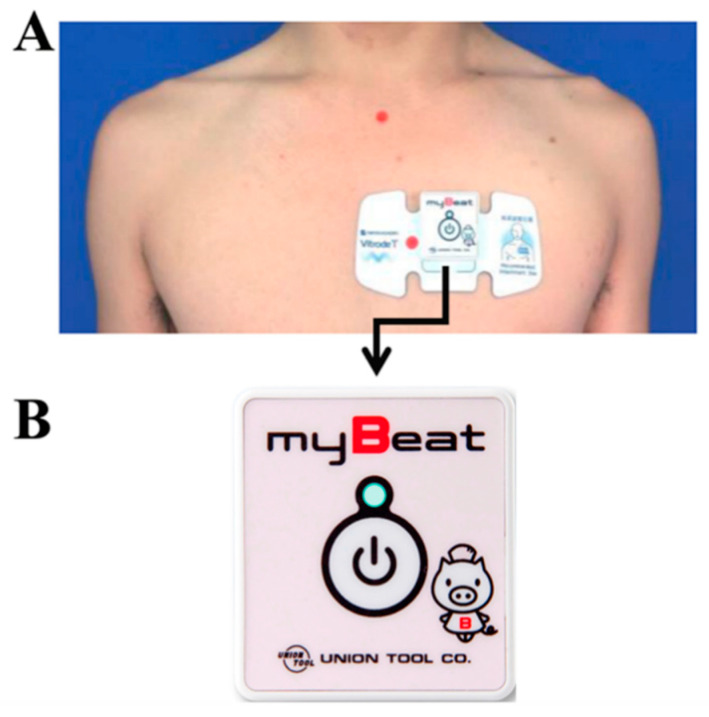
Wearable WHS-1 and electrode on the anterior chest. (**A**) Location of lead on the anterior chest. (**B**) Wearable WHS-1 apparatus.

**Figure 3 jcm-09-03359-f003:**
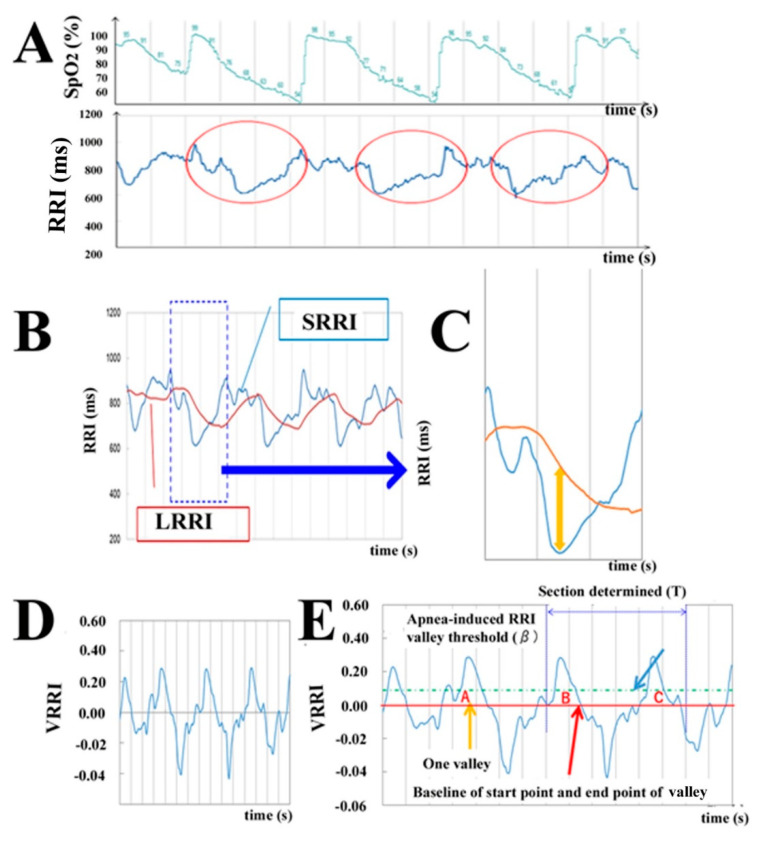
Analysis of the R-R interval (RRI). (**A**) The graphs of SpO_2_ and RRI when apnea and CVHR occur. The horizontal axis indicates time, and the vertical axis shows SpO_2_ (%, upper trace) and RRI (ms, lower trace). Note that the occurrence of apnea and CVHR (shown by red circles) is repeatedly observed. (**B**) The method of measuring the depth of the RRI valley to detect the deep valley of the RRI in CVHR. The long average of RRI (LRRI) and short average of RRI (SRRI) were calculated from the RRI of 60 s and 5 s, respectively (details in Methods). (**C**) The large-scale graph of SRRI and LRRI at the location where CVHR occurs. The horizontal axis indicates time (s), and the vertical axis indicates RRI (ms). It can be seen that, considering LRRI minus SRRI as the depth of the valley, the deep valley of RRI caused by CVHR indicated by the orange arrow can be identified. (**D**) Depth of the valley of the RRI fluctuation (VRRI) at a certain time (t) corrected by *LRRI*^2^ (details in Methods). (**E**) VRRI during apnea. The depth VRRI exceeds the threshold (≥0.090) at three points: (**A**–**C**), in this case.

**Figure 4 jcm-09-03359-f004:**
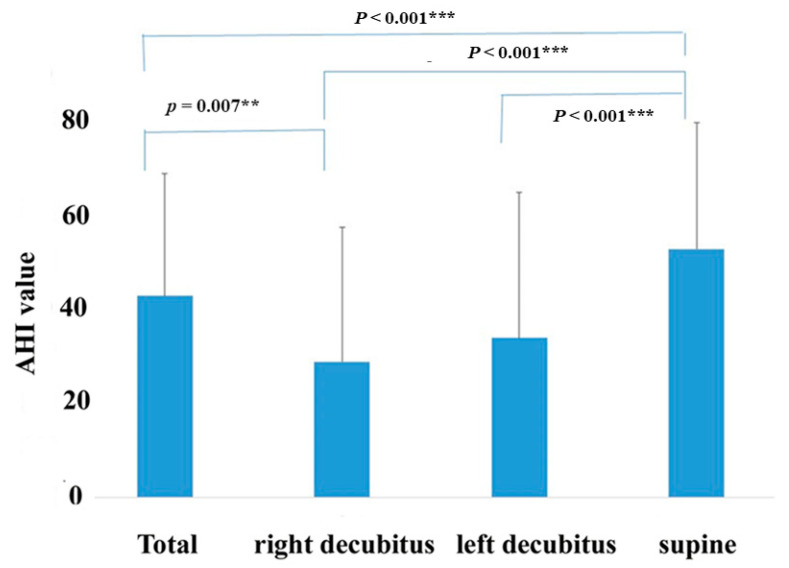
Change of the apnea-hypopnea index (AHI) by body position. The AHI value is shown in total periods, right lateral and left lateral decubitus, and supine position. Correlations between each position are illustrated. ** *p* < 0.01, and *** *p* < 0.001.

**Figure 5 jcm-09-03359-f005:**
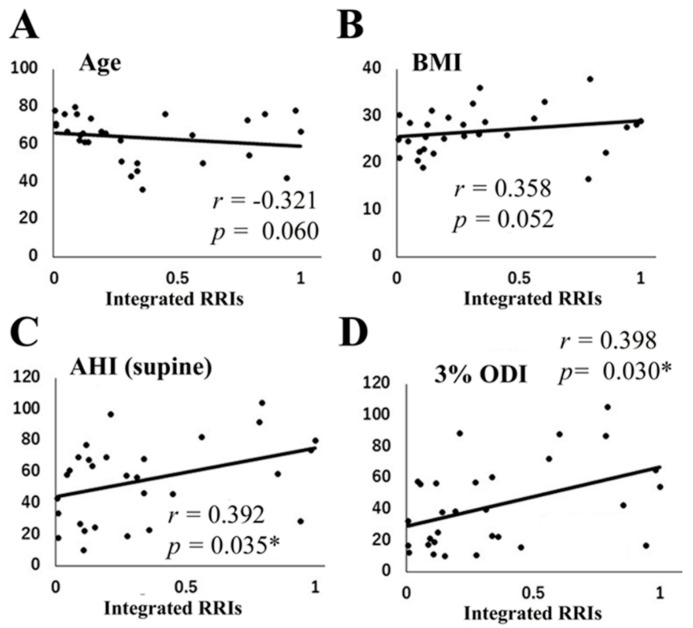
Correlation between various parameters and the ratio of integrated R-R intervals (RRIs). Correlations between the ratio of integrated RRIs and age (**A**), body mass index (BMI) (**B**), AHI (supine position, (**C**)), and 3% oxygen desaturation index (ODI) (**D**). * *p* < 0.05.

**Table 1 jcm-09-03359-t001:** Patient characteristics.

Number	30
Male: Female	22:8
Age, year	63.6 ± 12.1
BMI, kg/m^2^	26.8 ± 4.8
Risk factors, number	
Hypertension	19
Diabetes	7
Dyslipidemia	13
Smoking	18
Hyperuricemia	3
Hemodialysis	1
COPD	1
Cardiovascular disease, number	
CHF (HFpEF, HHD, OMI, Takotsubo)	4
pMVR	1
OMI	2
AAD	1
Drugs, number	
β-blockers	6
Ca-blockers	16
ACE-I/ARB	10
Diuretics	6
Statin	10
Oral diabetic drugs	6
Insulin	3

The mean ± SD values are shown. Body mass index, BMI; congestive heart failure, COPD; chronic obstructive pulmonary disease, CHF; heart failure with preserved ejection fraction, HFpEF; hypertensive heart disease, HHD; old myocardial infarction, OMI; pMVR; post-mitral valvuloplasty; acute aortic dissection, AAD; angiotensin converting enzyme inhibitor, ACE1, and angiotensin II receptor blocker, ARB.

**Table 2 jcm-09-03359-t002:** Correlation matrix between various parameters and the apnea-hypopnea index (AHI).

Parameters	Actual Data	*r*-Value (*p*-Value)
Age	63.6 ±11.6	−0.059 (0.725)
BMI	26.3 ± 4.6	0.532 (<0.001) **
Snoring rate (%)	29.7 ± 20.5	0.104 (0.539)
3% ODI	38.7 ± 26.2	0.987 (<0.001) **
Mean SpO_2_	95.7 ± 1.6	−0.399 (0.013) *
Minimum SpO_2_	78.7 ± 13.1	−0.652 (< 0.001) ***
Arousal Index	41.0 ± 22.2	0.800 (<0.001) ***
PLMS Index	9.4 ± 17.7	0.269 (0.121)

BMI, body mass index; ODI, the oxygen desaturation index; and PLMS, periodic leg movements (PLM) during sleep. * *p* < 0.05, ** *p* < 0.01, and *** *p* < 0.001.

**Table 3 jcm-09-03359-t003:** Correlation matrix between various parameters and integrated R-R intervals (RRIs).

Parameters	*r*-Value (*p*-Value)
Age	−0.321 (0.060)
BMI	0.358 (0.052)
Snoring rate (%)	0.111 (0.567)
AHI	0.416 (0.022) *
AHI (supine)	0.392 (0.035) *
AI	0.300 (0.108)
HI	0.097 (0.609)
3% ODI	0.398 (0.030) *
Mean SpO_2_	0.089 (0.641)
Minimum SpO_2_	−0.200 (0.298)
Arousal Index	0.252 (0.179)
PLMS Index	0.294 (0.122)

BMI, body mass index; ODI, oxygen desaturation index; AHI, apnea–hypopnea index; AI, apnea index; HI, hypopnea index; and PLMS, periodic leg movements during sleep. * *p* < 0.05.

**Table 4 jcm-09-03359-t004:** Multiple linear regression analysis of the integrated RRIs and AHI or 3% oxygen desaturation index (ODI).

	Dependent Variable: Log (Integrated RRIs)			
	Model 1	Model 2	Model 3	Model 4
Independent variable.	β-value (*p*-value)	β-value (*p*-value)	β-value (*p*-value)	β-value (*p*-value)
AHI (log)	0.382 (0.037) *	0.392 (0.023) *	0.464 (0.010) *	0.663 (0.003) **
	Model 1	Model 2	Model 3	Model 4
Independent variable.	β-value (*p*-value)	β-value (*p*-value)	β-value (*p*-value)	β-value (*p*-value)
3% ODI	0.381 (0.038) *	0.379 (0.028) *	0.464 (0.011) *	0.637 (0.008) **

Model 1, unadjusted; Model 2, adjusted by age; Model 3, adjusted by age and sex; and Model 4, age, sex, and BMI. * *p* < 0.05 and ** *p* < 0.01.

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
