# Peer review of "Clinical Usefulness of New R-R Interval Analysis Using the Wearable Heart Rate Sensor WHS-1 to Identify Obstructive Sleep Apnea: OSA and RRI Analysis Using a Wearable Heartbeat Sensor"

_jcm, 2020, doi:10.3390/jcm9103359_

Round 1
Reviewer 1 Report
This is a very fascinating study—congratulation on outstanding work. However, I have a few comments and suggestions.
- Not clear why you are using SAS and alternating between SAS and OSAS. It will be better to be consistent with one.
- Line 48 , Consider an updated reference for OSA; incidences are much higher now J Clin Sleep Med. 2019 Feb 15; 15(2): 301–334.
- Line 94 , need to clarify, was that just overnight oximetry, not sleep study.?
- Line 103-105, are these the criteria you are utilizing to score OSA in your institution? These are different from the AASM criteria.
- I agree with your assessment of the limitations of your study. Besides, one should pose the question of, what is the clinical value of WHS-1 or any other screening tool compared to the sleep questionnaires and clinical assessment of the physician. Unfortunately, such a device will not eliminate the need for further diagnosed studies ( PSG) to pursue medical management.
Thank you
Author Response
Reply to Reviewer #1
We greatly appreciate your careful attention to our manuscript and especially your excellent suggestions for improving the clarity and correctness of the message. We have corrected the paper as per your suggestions, and consider the revised manuscript much improved.
Reviewer #1:
This is a very fascinating study—congratulation on outstanding work. However, I have a few comments and suggestions.
1.Not clear why you are using SAS and alternating between SAS and OSAS. It will be better to be consistent with one.
#) Answer: Thank you very much for your suggestion. We used OSA only.
2.Line 48, Consider an updated reference for OSA; incidences are much higher now J Clin Sleep Med. 2019 Feb 15; 15(2): 301–334.
#) Answer: Thank you very much. I cited the following paper, and added the following sentence.
Patil SP, Ayappa IA, Caples SM, Kimoff RJ, Patel SR, Harrod CG. Treatment of Adult Obstructive Sleep Apnea with Positive Airway Pressure: An American Academy of Sleep Medicine Systematic Review, Meta-Analysis, and GRADE Assessment. J Clin Sleep Med. 2019 Feb 15;15(2):301-334. doi: 10.5664/jcsm.7638.
Page 2 line 47~
Recent data from the Wisconsin Sleep Cohort estimate that 17% of men and 9% of women aged 50 to 70 years have moderate to severe OAS [14,15].
3.Line 94, need to clarify, was that just overnight oximetry, not sleep study.?
#) Answer: You are right. We corrected it.
Overnight pulse oximetry was performed----.
4.Line 103-105, are these the criteria you are utilizing to score OSA in your institution? These are different from the AASM criteria.
#) Answer: Thank you very much for your suggestion. We used the American Academy of Sleep Medicine (AASM) recommendation (2014) as a scoring of hypopnea. I corrected it as follows.
Page 3 line 101~
The American Academy of Sleep Medicine (AASM) recommendation (2014) was used for a scoring of hypopnea [23]. We scored a respiratory event as a hypopnea if all of the following criteria are met. 1) The peak signal excursions drop by ≥30% of pre-event baseline. 2) The duration of the ≥30% drop in signal excursion is ≥10 s. 3) There is a ≥3% oxygen desaturation from pre-event baseline or the event is associated with an arousal. A reduction greater than 80% with the duration ≥10 s was defined as apnea.
- I agree with your assessment of the limitations of your study. Besides, one should pose the question of, what is the clinical value of WHS-1 or any other screening tool compared to the sleep questionnaires and clinical assessment of the physician. Unfortunately, such a device will not eliminate the need for further diagnosed studies (PSG) to pursue medical management.
#) Answer: Thank you very much for your comments. I absolutely agree with you. I added the following sentence in discussion.
Page 13 line 423~
Also, the additional studies are required to compare the usefulness for detecting OSA, by using WHS-1 system and other screening tool such as the sleep questionnaires and clinical assessment of the physician.

Reviewer 2 Report
The need to develop expeditious and cost-effective tools to screen Obstructive Sleep Apnea (OSA), especially in certain groups of patients remains a challenge and an issue worthy of on-going investigation. In this study the authors have looked at the utility of New R-R Interval Analysis Using the Wearable Heart Rate Sensor WHS-1 for OSA diagnosis. The study was performed on 30 patients in total. While the concept of this manuscript is important, there are several limitations that need to be addressed.
Major Comments
- The aim of the study is confusing please rephrase. The authors should not report conclusions in this section.
- In the methods section the authors should refine exactly how they scored respiration events, especially hypopneas. Given the AHI is the gold standard measurement it must be absolutely clear how it was scored as different scoring for hypopneas change the AHI substantively.
- The results section is a bit confused and should be more focused.
- Important limitations apart from the small study population is that, because of the open invitation for participation in the study and the lack of randomization of patients, patients with more frequent symptoms and greater concern about having OSA may have accepted the invitation, which would explain the high prevalence of severe OSA in this sample.
- Most of the text is rather confusing. In general the authors should structure their text so that each component makes sense on its own and also contributes to the main topic of the paper.
- Discussion section should be improved. The authors should discuss more the clinical implications of their findings. Furthermore, the predictive performance of the method could vary among different patient populations. Please comment.
- Please update reference list especially references associated with OSA prevalence
Author Response
Reply to Reviewer #2
We greatly appreciate your careful attention to our manuscript and especially your excellent suggestions for improving the clarity and correctness of the message. We have corrected the paper as per your suggestions, and consider the revised manuscript much improved.
The need to develop expeditious and cost-effective tools to screen Obstructive Sleep Apnea (OSA), especially in certain groups of patients remains a challenge and an issue worthy of on-going investigation. In this study the authors have looked at the utility of New R-R Interval Analysis Using the Wearable Heart Rate Sensor WHS-1 for OSA diagnosis. The study was performed on 30 patients in total. While the concept of this manuscript is important, there are several limitations that need to be addressed.
Major Comments
1.The aim of the study is confusing please rephrase. The authors should not report conclusions in this section.
#) Answer: Thank you very much for your suggestion. We added the following sentenses.
Page 2 line 66~
The aim of the present study is to investigated the usefulness of newly developed RRI analysis using the mobile and wearable heart rate sensor WHS-1 to identify patients with OSA.
Page 14 line 429~
We have developed the RRI analysis using the wearable heart rate sensor WHS-1 and new algorithm, which may become an expeditious and cost-effective screening tool for identifying OSA.
- In the methods section the authors should refine exactly how they scored respiration events, especially hypopneas. Given the AHI is the gold standard measurement it must be absolutely clear how it was scored as different scoring for hypopneas change the AHI substantively.
#) Answer: Thank you very much for your comments. We used the American Academy of Sleep Medicine (AASM) recommendation (2014) as a scoring of hypopnea. I corrected it as flows.
Page 3 line 101~
The American Academy of Sleep Medicine (AASM) recommendation (2014) was used for a scoring of hypopnea [23]. We scored a respiratory event as a hypopnea if all of the following criteria are met. 1) The peak signal excursions drop by ≥30% of pre-event baseline. 2) The duration of the ≥30% drop in signal excursion is ≥10 s. 3) There is a ≥3% oxygen desaturation from pre-event baseline or the event is associated with an arousal. A reduction greater than 80% with the duration ≥10 s was defined as apnea.
3.The results section is a bit confused and should be more focused.
#) Answer: I have re-arranged it. Thank you very much.
1.Important limitations apart from the small study population is that, because of the open invitation for participation in the study and the lack of randomization of patients, patients with more frequent symptoms and greater concern about having OSA may have accepted the invitation, which would explain the high prevalence of severe OSA in this sample.
#) Answer: You are absolutely right. I added it in discussion.
Page 13 line 418~
We have not applied this method to different populations, particularly those with a low pretest probability of OSA. In addition, given the needs for screening of OSA in general practice and health promotion, studies in the general population are also very important. Thus, further studies using an open invitation for participation in the study and randomization of patients including low-risk OSA are required to clarify the clinical usefulness of our program with a wearable heart rate sensor apparatus.
5.Most of the text is rather confusing. In general, the authors should structure their text so that each component makes sense on its own and also contributes to the main topic of the paper.
#) Answer: Thank you very much for comments. You are absolutely right.
The main topic of the paper is to show the newly developed RRI analysis using the wearable heart rate sensor WHS-1 and new algorithm. The method may become an expeditious and cost-effective screening tool for identifying SAS. I re-arranged it.
1.Discussion section should be improved. The authors should discuss more the clinical implications of their findings. Furthermore, the predictive performance of the method could vary among different patient populations. Please comment.
#) Answer: Thank you very much for your comments. You are right. Our study shows that the newly- developed - RRI analysis using the wearable heart rate sensor WHS-1 may become an expeditious and cost-effective screening tool for identifying OSA. However, as your suggestion, important limitations apart from the small study population is that, because of the open invitation for participation in the study and the lack of randomization of patients, patients with more frequent symptoms and greater concern about having OSA may have accepted the invitation, which would explain the high prevalence of severe OSA in this sample.
Therefore, I commented about it in discussion.
Page 13 line 416~
There are several limitations in our current study. First, we studied a small number of patients with OSA who underwent a PSG sleep study, and who could be diagnosed with OSA by PSG findings. We have not applied this method to different populations, particularly those with a low pretest probability of OSA. In addition, given the needs for screening of OSA in general practice and health promotion, studies in the general population are also very important. Thus, further studies using an open invitation for participation in the study and randomization of patients including low-risk OSA are required to clarify the clinical usefulness of our program with a wearable heart rate sensor apparatus. Also, the additional studies are required to compare the usefulness for detecting OSA, by using WHS-1 system and other screening tool such as the sleep questionnaires and clinical assessment of the physician.
2.Please update reference list especially references associated with OSA prevalence
#) Answer: I updated OSA prevalence lists as the referee 1 suggestion.
- Patil, S,P.; Ayappa, I.A.; Caples, S.M.; Kimoff, R.J.; Patel, S.R.; Harrod, C.G. Treatment of Adult Obstructive Sleep Apnea With Positive Airway Pressure: An American Academy of Sleep Medicine Systematic Review, Meta-Analysis, and GRADE Assessment. J Clin Sleep Med. 2019, 15, 301-334.
- Peppard, P.; Young, T.; Barnet, J.H.; Palta, M.; Hagen, E.W.; Hla, K.M. Increased Prevalence of Sleep-Disordered Breathing in Adults. Am J Epidemiol. 2013,177, 1006-1014.

Round 2
Reviewer 2 Report
The authors have sufficiently answered to reviewer's comments.
Minor comments
Language improvements are suggested